# Postoperative Acute-Phase Gait Training Using Hybrid Assistive Limb Improves Gait Ataxia in a Patient with Intradural Spinal Cord Compression Due to Spinal Tumors

**DOI:** 10.3390/medicina58121825

**Published:** 2022-12-12

**Authors:** Yuichiro Soma, Shigeki Kubota, Hideki Kadone, Yukiyo Shimizu, Yasushi Hada, Masao Koda, Yoshiyuki Sankai, Masashi Yamazaki

**Affiliations:** 1Department of Rehabilitation Medicine, University of Tsukuba, Tsukuba 305-8576, Ibaraki, Japan; 2Department of Orthopaedic Surgery, Faculty of Medicine, University of Tsukuba, Tsukuba 305-8576, Ibaraki, Japan; 3Center for Innovating Medicine and Engineering (CIME), University of Tsukuba, Tsukuba 305-8576, Ibaraki, Japan; 4Faculty of Systems and Information Engineering, University of Tsukuba, Tsukuba 305-0006, Ibaraki, Japan

**Keywords:** hybrid assistive limb, robotic rehabilitation, non-traumatic spinal cord dysfunction, gait ataxia, electromyography, motion analysis

## Abstract

Sensory ataxia due to posterior cord syndrome is a relevant, disabling condition in nontraumatic spinal cord dysfunction. Ataxic gait is a common symptom of sensory ataxia that restricts activities of daily living. A 70-year-old woman with severe sensory disturbance was diagnosed with intradural extramedullary spinal cord tumors found in the thoracic spine region (T8). Surgical management of the tumors was performed. The patient received gait training 20 days after surgery (postoperative acute phase) using a hybrid assistive limb (HAL). HAL is a wearable exoskeleton cyborg that provides real-time assistance to an individual for walking and limb movements through actuators mounted on the bilateral hip and knee joints. Walking ability was assessed using the 10 m walking test, which included evaluating walking speed, step length, and cadence in every session. To evaluate the immediate effects of HAL training, walking speed and step length were measured before and after the training in each session. During the 10 m walking test, gait kinematics and lower muscle activity were recorded using a motion capture system and wireless surface electromyography before the first session and after completion of all HAL sessions. After the HAL training sessions, improvement in the patient’s gait performance was observed in the gait joint angles and muscle activity of the lower limb. After 10 training sessions, we observed the following changes from baseline: walking speed (from 0.16 m/s to 0.3 m/s), step length (from 0.19 m to 0.37 m), and cadence (from 50.9 steps/min to 49.1 steps/min). The average standard deviations of the knee (from right, 7.31; left, 6.75; to right, 2.93; *p* < 0.01, left, 2.63; *p* < 0.01) and ankle joints (from right, 6.98; left, 5.40; to right, 2.39; *p* < 0.01, left, 2.18; *p* < 0.01) were significantly decreased. Additionally, walking speed and step length improved immediately after completing all the HAL training sessions. This suggests that HAL gait training might be a suitable physical rehabilitation program for patients with sensory ataxia causing dysfunctional movement of the lower limb.

## 1. Introduction

Non-traumatic spinal cord dysfunction is thought to be more common than traumatic spinal cord injury in many developed countries [1], and one of its main causes is tumors of the spinal cord [2]. Several reports have described the benefits of rehabilitation in specialized facilities for patients with non-traumatic spinal cord dysfunction [3,4,5]. The primary goal of physical rehabilitation of patients with this condition is the maximum possible recovery of functional independence and reintegration into daily life [6]. Spinal cord compression by tumors can cause posterior cord syndrome, which is described as a selective dysfunction of the posterior columns of the spinal cord, resulting in sensory ataxia [7]. Despite the preservation of motor functional strength, the loss of proprioception and position sense in posterior cord syndrome can lead to a significant decrease in mobility and self-care abilities. We believe these complications can hinder the progression of rehabilitation in patients with non-traumatic spinal cord dysfunction, as there is a lack of evidence regarding the most convenient rehabilitation for sensory ataxia.

A hybrid assistive limb (HAL) robot is a wearable exoskeleton cyborg that provides real-time assistance to an individual for walking and performing limb movements through actuators mounted on the bilateral hip and knee joints [8]. To date, no study has been conducted on HAL training for patients with sensory ataxia caused by non-traumatic spinal cord dysfunction. Importantly, restoration of motor functions induces structural reorganization of neuronal circuits in the injured spinal cord [9]. A previous study reported that HAL training improved gait speed and endurance in patients with spinal cord injury [10]. However, spinal cord injury may not be comparable with non-traumatic spinal cord dysfunction since different functions may be impaired. For example, ataxic gait due to loss of proprioception in posterior cord syndrome requires sufficient supraspinal input from the peripheral nervous system. We hypothesized that HAL training can improve the walking ability through voluntary drive and normalized motion assistance provided by the external device, thus forming a foundation for a proprioceptive feedback loop for patients with lesions involving sensory pathways [11]. The aim of this study was to show the feasibility and efficacy of HAL training for a patient with sensory ataxia caused by an intradural spinal cord tumor by assessing its effect on walking abilities in terms of gait kinematics and kinetics to show that HAL gait training could be a beneficial rehabilitation program.

## 2. Case Presentation

### 2.1. Patient Information

A 70-year-old woman had been increasingly facing walking difficulties for approximately 3 months. When her walking impairment started, she experienced several physical problems, including flank pain, balance impairment, sensory deficits, and muscle weakness (manual muscle testing grading system 3) under the umbilical region. She was diagnosed with intradural extramedullary spinal cord meningioma, which was observed in the thoracic spine region (T8) using T2-weighted magnetic resonance imaging and computed tomography images (Figure 1A–C). Surgical management of the tumor was planned. The day before the surgery, a physiotherapist evaluated the patient’s neurological impairment and disability. The patient could perform only mild voluntary movements in both lower limbs; showed sensory disorders, including hypesthesia of the hip, anus, and lower limbs; and required assistance for activities such as standing and walking, indicating severe impairment of activities of daily living.

### 2.2. Surgical Procedure and Postoperative Courset

Surgical treatment involved T7–T8 laminectomy, T9 partial laminectomy, and tumor resection. The dura mater was cut transversely in the ventral direction, and the tumor and dura mater were removed in a lump. All surgical procedures were undertaken carefully and accomplished uneventfully. The tumor was diagnosed as a transitional meningioma using histopathology (normal cellularity, sheeting, small cell, prominent nucleoli, spinal cord invasion, and few mitoses; World Health Organization classification grade I). The day after surgery, the patient was allowed to sit and walk without an orthosis. Conventional rehabilitation was initiated immediately after permission by the attending physician, since the patient had severe dysfunctional control of voluntary movements and balance and was unable to stand and walk without assistance.

### 2.3. Clinical Findings

A week after surgery, the patient’s physical condition showed a neurological level of injury (NLI) of T6, American Spinal Cord Injury Association (ASIA) impairment scale (AIS) grade C [12], International Standards for Neurological and Functional Classification of Spinal Cord Injury (ISNCSCI) motor score [12,13] of 83 points (right: 46 points; left: 37 points), sensory score of 71 points for light touch (right: 32 points; left: 39 points) and 61 points for a pin prick (right: 27 points; left: 34 points), and walking index for spinal cord injury (WISCI) II score of 1 point [14,15]. The modified Ashworth scale of the ankle plantar flexors was 2.5 points [16]. Her Berg Balance Scale (BBS) [17] and Functional Independence Measure (FIM) motor scores [18] were 3 and 37 points, respectively.

## 3. Methods

### 3.1. Measurement

We evaluated the occurrence of adverse events related to HAL training. For that, we evaluated some variables, including walking speed, step length, and cadence, using the 10 m walking test before the first HAL training session (before 1 session, baseline) and after completion of the entire set of sessions (10 sessions). The NLI, ASIA impairment scale classification, ISNCSCI motor and sensory scores, WISCI-II, BBS, FIM, and modified Ashworth scale scores for the lower extremity were also assessed [12,13,14,15,16,17,18]. Walking speed and step length were measured with the 10 m walking test before (pre-HAL) and after (post-HAL) HAL training in each session to evaluate the immediate effect of HAL training. The patient was instructed to walk on a flat surface at a comfortable pace during the 10 m walking test, and walking time was measured using a handheld stopwatch, which was used to calculate the walking speed. The number of steps taken between the start and finish line was counted to calculate the step length (meters), and the cadence was calculated based on the number of steps taken over the walking time, measured as the number of steps per minute.

### 3.2. Joint Angles

To evaluate whether the HAL intervention affects lower limb movement and gait kinematics, the 10 m walking test was conducted. The measurements recorded without the participant wearing the HAL suit at the baseline and after the 10 HAL training sessions were compared using a motion capture system (Vicon MX with 16 T20S cameras, 100 Hz, Plug-in gait marker set, Oxford, UK) [19,20,21]. Briefly, according to a plug-in gait marker set, auto-reflective markers were placed on the following anatomical landmarks: anterior superior iliac spine, posterior superior iliac spine, lower lateral one-third surface of the thigh, flexion-extension axis of the knee, lower lateral one-third surface of the shank, head of the second metatarsal bone of the toe, lateral malleolus of the ankle, and posterior peak of the heel calcaneus.

### 3.3. Electromyography

Electromyography (EMG) was recorded during the 10 m walking test without the participant wearing the HAL suit at the baseline and after 10 sessions. Additionally, we recorded and compared EMG activity profiles of lower limb muscles during the first and tenth HAL training sessions with and without HAL. As previously described [17,18,19], EMG sensors were placed bilaterally on the muscle relevant to the hip, knee, and ankle joints. Muscle activity was recorded for the rectus femoris, gluteus maximus, hamstrings, tibialis anterior, and gastrocnemius muscles using the TrignoTM Lab Wireless EMG System (Delsys, Inc., Boston, MA, USA) sampling at 2 kHz synchronized with the motion capture system.

### 3.4. Therapeutic Intervention

#### HAL Gait Training

HAL gait training was initiated 20 days after surgery and included a total of 10 sessions over 1 month, with 2–3 sessions per week. During this period, the patient continued her conventional rehabilitation performed by a physiotherapist and occupational therapist four times a week. Two therapists and an assistant attached and detached the HAL exoskeleton suit, and an engineer performed the gait movement analysis. All HAL training sessions, including standing, sitting, and walking, were performed with assistance from the All-in-One Walking Trainer (All-in-One Walking Trainer, Ropox A/S, Naestved, Denmark) to prevent falls during the training; body weight support was provided using a harness. HAL training was performed with adjustable body weight support under the supervision of a physiotherapist. During all training sessions, the patient used an ankle foot orthosis to prevent ankle sprain due to lower-extremity ataxia. The patient had severe sensory disturbance; therefore, it was important to gain proprioceptive feedback in her foot and achieve appropriate foot movement. Our aim was to provide the patient with a patient-appropriate sensory input with her own promotion on the ground. A previous study suggested that advanced robotic technology, such as the EksoTM, allows users to actively participate in walking control through subtle trunk motions, which help in shifting their weight over the appropriate foot to trigger each step on the ground [22]. Therefore, in this case, we conducted the interventional HAL gait training on the ground and not on a treadmill. The duration of one training session was approximately 60 min, including both fitting and evaluation. The net gait training time was 15–20 min. This case primarily used the cybernic voluntary control mode of the HAL exoskeleton, which allows the operator to adjust the degree of physical support provided for the patient’s comfort. Walking distance, walking speed, and level of body weight support during the HAL training sessions were determined considering the patient’s vital signs and assessing her fatigue and motivation.

### 3.5. Data Analysis

#### Joint Angles

For each cycle, the maximum and minimum angles were extracted at the hip, knee, and ankle joints or the maximum anterior and posterior angles of the pelvis to calculate the joint range of motion. The kinematic profile of each lower extremity for each extracted step cycle was normalized to the duration of the step cycle and averaged across cycles. The extraction process also analyzed the standard deviations from the average of each joint angle and pelvis angles. The average standard deviations at baseline and after 10 sessions were compared using a *t*-test. Statistical significance was set at *p* < 0.05.

### 3.6. Electromyography

EMG data were filtered with a 30–400 Hz bandwidth passing filter using scripts on MATLAB 8.2 (MathWorks, Natick, MA, USA). The data were rectified and locally integrated using a 200 ms moving window to obtain an integrated EMG profile and then time-normalized to 100 time points.

## 4. Results

No serious adverse events related to HAL training were observed in any session. The immediate effects on walking speed and step length are shown in Figure 1A. The 10 m walking test showed changes in walking speed (from 0.16 m/s at baseline to 0.3 m/s after 10 sessions), step length (from 0.19 m at baseline to 0.37 m after 10 sessions), and cadence (from 50.9 steps/min at baseline to 49.1 steps/min after 10 sessions) (Figure 1B, Appendix A). The patient’s gait performance in the double stance phase immediately before each HAL gait training session showed that some sagittal postures had improved (Figure 1B).

The sessions greatly improved the walking speed and step length. The maximum extension angles of the hip joint during the stance phase increased from −28.9° at the baseline to 7.5° after 10 sessions and −28.3° at baseline to 0.5° after 10 sessions, on the left and right sides, respectively. The maximum extension angles of the knee joints after 10 sessions also increased from −18.6° to 4.91° and from −15.9° to 3.4° on the left and right side, respectively. The maximum dorsiflexion angles of the ankle joint during the swing phase increased from −2.83° at baseline to 13.5° after 10 sessions on the left and decreased from 23.0° at baseline during the swing phase to 15.4° after 10 sessions during the stance phase on the right (Figure 2A). The average standard deviations of the knee (baseline, right, 7.31; left, 6.75; after 10 sessions, right, 2.93; *p* < 0.01, left, 2.63; *p* < 0.01) and ankle joints (baseline, right, 6.98; left, 5.40; after 10 sessions, right, 2.39; *p* < 0.01, left, 2.18; *p* < 0.01) were significantly decreased. The average of the standard deviations of the pelvic angles was significantly increased (baseline, −1.45; after 10 sessions, 24.29; *p* < 0.01). However, no significant difference in the average standard deviations of the hip joints was observed (baseline, right, 2.38; left, 3.34; after 10 sessions, right, 2.79; *p* < 0.03589, left, 3.23; *p* < 0.5926) (Figure 2B). Lower limb muscle activation during the gait cycle was rarely observed before HAL gait training. However, after HAL gait training, lower limb muscle activation during the gait cycle increased (Figure 2C).

Figure 3 shows the band-pass filtered EMG data for the rectus femoris and gluteus maximus muscles during HAL training, with and without HAL in the first and tenth sessions. In the first session, the activation of both the muscles in the stance phase was increased during training with HAL (Figure 3A), whereas in the 10th session, activation of both the muscles increased during training, both with and without HAL (Figure 3B). Other outcomes, including the ISNSCI motor score (from 83 to 93 points), ISNSCI sensory score for light touch (from 71 to 89 points) and a pin prick (from 61 to 91 points), WISCI-II score (from 1 to 6, with progression from using a walker and ankle foot orthosis), BBS (from 3 to 12 points, sitting to standing, sitting unsupported, standing to sitting, and transfers), FIM motor score (from 37 to 64 points), and modified Ashworth scale of ankle plantar flexors (from 2.5 to 1 points), also showed improvement. The NLI changed from T6 to T9, and the ASIA impairment scale classification changed from grade C to D. A long-term follow-up was implemented after the HAL gait training.

## 5. Discussion

HAL gait training was conducted in a patient with non-traumatic spinal cord dysfunction due to spinal cord compression by spinal tumors. During training, her gait profiles gradually changed in terms of gait kinematics and kinetics, specifically the maximum extension angles of the hip and knee joints during the stance phase and the dispersion characteristics of the pelvis, knee, and ankle angles. Furthermore, each EMG muscle response was activated after HAL training. Therefore, we believe that the patient’s gait performance was close to normal [23] after the training sessions.

To our knowledge, this is a relatively rare case report describing the feasibility and safety of HAL training in patients with posterior cord syndrome caused by non-traumatic spinal cord dysfunction. As conventional rehabilitation was started after surgery in our patient, the lower-extremity ataxia caused severe balance impairment and hindered the possibility of continuing daily living activities. Thus, it is likely that HAL training enhanced muscle movement or strength, therefore improving motor coordination and balance impairment.

Neural activity and repeated execution of specific tasks promote and lead to the reinstatement or restructuring of appropriate proprioceptive feedback learning [24]. The improvement in gait speed and step length after HAL training sessions is speculated as follows. Despite having moderate muscle strength (AIS C) before the HAL training session, the patient did not have normal efficient joint movement and muscle activity for normal mobilized gait pattern. In the first session, the rectus femoris and gluteus maximus were activated to a greater extent during HAL gait training than during conventional gait performance without HAL. In the 10th session, even during training without HAL, both muscles were activated, implying that the patient reintegrated the motions of the hip and knee joint muscle response through motor learning with HAL’s motion support [25]. However, we evaluated the dispersion characteristics of the pelvis, hip, knee, and ankle angles for the impairment of lower limb mobility due to sensory ataxia. The pelvic forward tilting angle increased significantly after the HAL training, while the knee and ankle dispersion angles decreased significantly. The results indicate that HAL training might help in correcting the patient’s abnormal gait pattern, leading to gains in motor learning and ensuring consistent signaling of appropriate sensorimotor interactions in the patient’s central nervous system. A previous study reported increased muscle synergies while walking with HAL, suggesting that spinal gait control is altered with HAL [19]. Another study reported that HAL motion assistive technologies could contribute to improvements in walking abilities by facilitating proper joint motion and loading and unloading muscle movements in patients with chronic severe incomplete tetraplegic SCI [26]. The current study conducted HAL training for a patient with non-traumatic spinal cord dysfunction; however, it is likely to observe similar improvements of gait profiles.

Moreover, we focused on the immediate effect of HAL on gait parameters. The parameters did not increase from a certain level at the pre-HAL stage to after three sessions in walking speed and after five sessions in step length. However, the post-HAL training results showed improvement in these parameters. Furthermore, the gap between the pre-(Figure 1A; blue line) and post-HAL (Figure 1A; red line) results gradually increased after five sessions compared to the gap before four sessions. We postulate that the patient became accustomed to HAL training, wherein repeated execution of specific tasks promoted the reinstatement or restructuring of appropriate proprioceptive feedback learning. In this case, HAL gait training appears to be a convenient rehabilitation program that corrects dysfunctional movement due to sensory ataxia, as it might not only affect somatic or sensory discrimination deficits but also has the potential to improve neuronal plasticity, leading to coordinated normal movements and contributing to motor learning.

This report has some limitations. Here, we highlighted the functional improvement observed with HAL gait training and focused on the effect of HAL on the central nervous system, but the interpretation of the results may seem excessive. Improvements in outcomes are expected in natural recovery after surgery or conventional rehabilitation training. Furthermore, to prove the effectiveness of HAL, it is necessary to show that the gain in gait parameters after HAL training exceeds the gain after conventional training. However, we analyzed only one patient. Additionally, we only analyzed the band-pass filtered EMG data of the rectus femoris and gluteus maximus muscles, which were compared during the first and tenth sessions of HAL training (with and without HAL). Muscle network and synergy analysis of the EMG may serve as a useful tool to evaluate muscle activity. Advances in these analyses of muscle activity and evaluation during each HAL training session may help further understanding of this issue. Another limitation of this study is that it is a report of a single patient. It is important to demonstrate the effectiveness of HAL in further studies, in a larger study with multiple patients and control groups.

## 6. Conclusions

To conclude, this study is the first to report the feasibility, safety, and efficacy of HAL training for patients with non-traumatic spinal cord dysfunction due to spinal cord compression by tumors. The present case suggests that HAL gait training might be a highly beneficial rehabilitation program for patients with sensory ataxia of the lower limb causing dysfunctional movement.

## Figures and Tables

**Figure 1 medicina-58-01825-f001:**
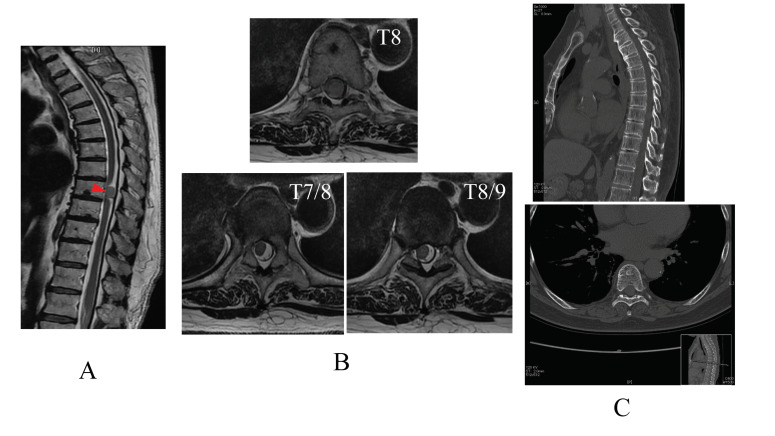
Mid-sagittal T2-weighted (T2WI) magnetic resonance image (MRI) of the thoracic spine region (T8, red arrow) (**A**) and coronal T2WI MRI scan, at the level of T7/8, T8, T8/9 (**B**). Sagittal and coronal computed tomography at the level of T8 (**C**). (**D**) Graph showing the chronological improvement in walking speed and step length in the 10 m walking test without the hybrid assistive limb (HAL) during the HAL gait sessions.

**Figure 2 medicina-58-01825-f002:**
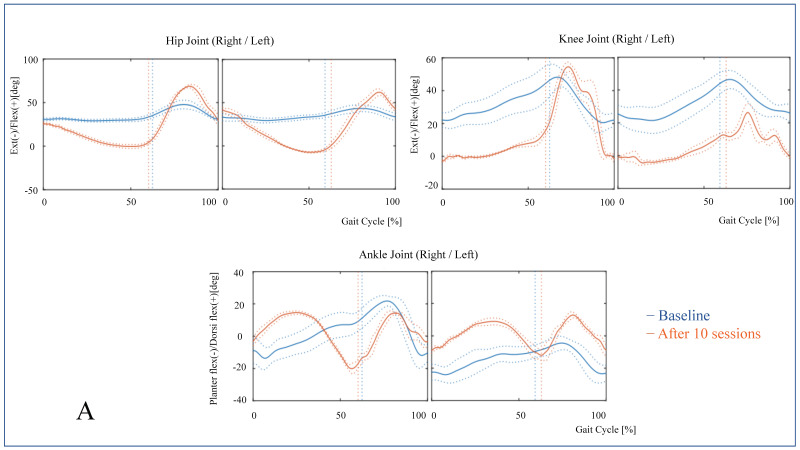
(**A**) Temporal profile of the angular position of the hip, knee, and ankle joints over the gait cycle, measured without the hybrid assistive limb (HAL) before session 1 (baseline) and after all 10 HAL training sessions. Vertical lines indicate the moment of toe lift. (**B**) Standard deviations from the average of the hip, knee, and ankle joint and pelvic angles. (**C**) Muscle activities of the rectus femoris, hamstrings, gastrocnemius, tibialis anterior, and gluteus maximus without the hybrid assistive limb (HAL) before session 1 (baseline) and after all 10 HAL training sessions. Each muscle response is activated during the gait cycle. The gastrocnemius muscles show greater response at the baseline than after 10 sessions, from the initial to the terminal during the stance phase. Vertical lines indicate the moment of toe lift.

**Figure 3 medicina-58-01825-f003:**
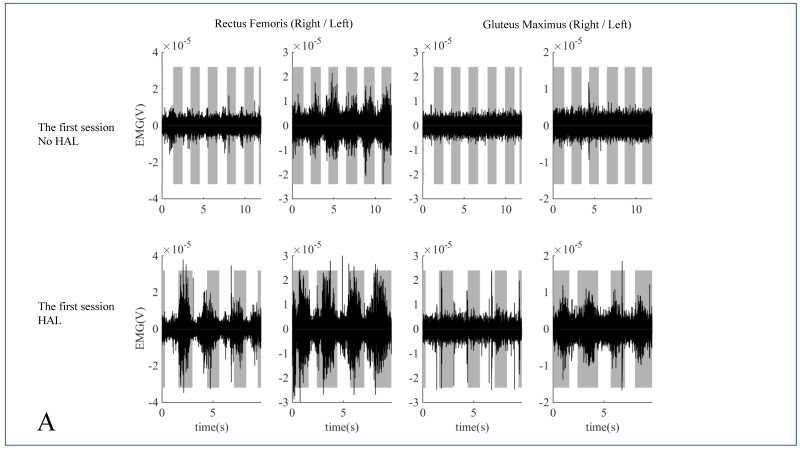
(**A**) Band-pass filtered EMG data of the rectus femoris and gluteus maximus muscles compared during the HAL training (with and without HAL) in the first session. (**B**) Band-pass filtered EMG data of the rectus femoris and gluteus maximus muscles compared during the HAL training (with and without HAL) in the tenth session.

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
