# Peer review of "Postoperative Acute-Phase Gait Training Using Hybrid Assistive Limb Improves Gait Ataxia in a Patient with Intradural Spinal Cord Compression Due to Spinal Tumors"

_medicina, 2022, doi:10.3390/medicina58121825_

Round 1

Reviewer 1 Report

This article represents case study regarding patient with posterior cord syndrome who received gait training after surgery using hybrid assistive limb. This is rather novel field with lack of previous research, however it has to be mentioned that this is only a case study.

These are my comments and suggestions:

Abstract:

Abstract is nicely written, however, I suggest to use quantitative data regarding patient's improvement.

Introduction:

Introduction includes relevant theoretical background and states the lack of previous research regarding this thematic. However, I advise to include a sentence with clear aim/goal of this study.

Case Presentation.

Please include the references for the validity and reliability of used tests. I suggest to divide this chapters into two separate chapters: Methods and Results.

Discussion:

Nicely written with clearly stated limitation and lack of previous research. If possible, compare your research with other studies using HAL.

Author Response

General comment
This article represents case study regarding patient with posterior cord syndrome who received gait training after surgery using hybrid assistive limb. This is rather novel field with lack of previous research, however it has to be mentioned that this is only a case study.

Response

Thank you for your comments and valuable remarks. We have ensured that all your concerns are addressed in the revised manuscript. Please find our responses to specific comments below.

Specific comment

Abstract: Abstract is nicely written, however, I suggest to use quantitative data regarding patient's improvement.

Response

Unfortunately, there is a strict word limit for the abstract which is why we focused on the main results only, i.e. the improvement of gait profiles. However, we have now revised the abstract to include more quantitative data.

Revised Manuscript in the abstraction section (Page: 1, lines: 27-32)

“Sensory ataxia due to posterior cord syndrome is a relevant, disabling condition in nontraumatic spinal cord dysfunction. Ataxic gait is a common symptom of sensory ataxia that restricts activities of daily living. A 70-year-old woman with severe sensory disturbance was diagnosed with intradural extramedullary spinal cord tumors found in the thoracic spine region (T8). Surgical management of the tumors was performed. The patient received gait training 20 days after surgery (postoperative acute phase) using a hybrid assistive limb (HAL). HAL is a wearable exoskeleton cyborg that provides real-time assistance to an individual for walking and limb movements through actuators mounted on the bilateral hip and knee joints. Walking ability was assessed using the 10-meter walking test, which included evaluating walking speed, step length, and cadence in every session. To evaluate the immediate effects of HAL training, walking speed and step length were measured before and after the training in each session. During the 10-meter walking test, gait kinematics and lower muscle activity were recorded using a motion capture system and wireless surface electromyography before the first session and after completion of all HAL sessions. After the HAL training sessions, improvement in the patient’s gait performance was observed in the gait joint angles and muscle activity of the lower limb. After 10 training sessions, we observed the following changes from baseline: walking speed (from 0.16 m/s to 0.3 m/s), step length (from 0.19 m to 0.37 m), and cadence (from 50.9 steps/min to 49.1 steps/min). The average standard deviations of the knee (from right, 7.31; left, 6.75; to right, 2.93; p < 0.01, left, 2.63; p < 0.01) and ankle joints (from right, 6.98; left, 5.40; to right, 2.39; p < 0.01, left, 2.18; p < 0.01) were significantly decreased. Additionally, walking speed and step length improved immediately after completing all the HAL training sessions. This suggests that HAL gait training might be a suitable physical rehabilitation program for patients with sensory ataxia causing dysfunctional movement of the lower limb. ”

Specific comment
Introduction: Introduction includes relevant theoretical background and states the lack of previous research regarding this thematic. However, I advise to include a sentence with clear aim/goal of this study.

Response

We fully agree and have added a corresponding statement at the end of the introduction section.

Revised Manuscript in the introduction section (Page: 2, lines: 68-72)

“We hypothesized that HAL training can improve the walking ability through voluntary drive and normalized motion assistance provided by the external device, thus forming a foundation for a proprioceptive feedback loop for patients with lesions involving sensory pathways [11]. The aim of this study was to show the feasibility and efficacy of HAL training for a patient with sensory ataxia caused by an intradural spinal cord tumor by assessing its effect on walking abilities in terms of gait kinematics and kinetics to show that HAL gait training could be a beneficial rehabilitation program.”

Specific comment

Case Presentation: Please include the references for the validity and reliability of used tests. I suggest to divide this chapters into two separate chapters: Methods and Results.

Response

Following your suggestion, we have included references for the validity and reliability of the tests and modified the chapters.

Revised Manuscript in the case presentation section (Page: 3, lines: 104-112)

A week after surgery, the patient’s physical condition showed a neurological level of injury (NLI) of T6, American Spinal Cord Injury Association (ASIA) impairment scale (AIS) grade C [12], International Standards for Neurological and Functional Classification of Spinal Cord Injury (ISNCSCI) motor score [12,13] of 83 points (right: 46 points; left: 37 points), sensory score of 71 points for light touch (right: 32 points; left: 39 points) and 61 points for a pin prick (right: 27 points; left: 34 points), and walking index for spinal cord injury (WISCI) II score of 1 point [14, 15]. The modified Ashworth scale of the ankle plantar flexors was 2.5 points [16]. Her Berg Balance Scale (BBS) [17] and Functional Independence Measure (FIM) motor scores [18] were 3 and 37 points, respectively.

  1. Methods (Page: 3, lines: 113)

Measurement

We evaluated the occurrence of adverse events related to HAL training. For that, we evaluated some variables, including walking speed, step length, and cadence using with the 10-meter walking test before the first HAL training session (before 1 session, baseline) and after completion of the entire set of sessions (10 sessions). The NLI, ASIA impairment scale classification, ISNCSCI motor and sensory scores, WISCI-II, BBS, FIM, and modified Ashworth scale scores for the lower extremity were also assessed [12-18]. Walking speed and step length were measured with the 10-meter walking test before (pre-HAL) and after (post-HAL) HAL training in each session to evaluate the immediate effect of HAL training. The patient was instructed to walk on a flat surface at a comfortable pace during the 10-meter walking test, and walking time was measured using a handheld stopwatch, which was used to calculate the walking speed. The number of steps taken between the start and finish line was counted to calculate the step length (meters) and the cadence was calculated based on the number of steps taken over the walking time, measured as the number of steps per minute.

  1. Results (Page: 5, lines: 197)

No serious adverse events related to HAL training were observed in any session. ... the ASIA impairment scale classification changed from grade C to D. A long-term follow-up was implemented after the HAL gait training.

Specific comment

Discussion:

Nicely written with clearly stated limitation and lack of previous research. If possible, compare your research with other studies using HAL.

Response

Since HAL is a relatively new technique, only a few studies have used it so far. We included one additional study in the discussion section.

Revised Manuscript in the discussion section (Page: 7, lines: 291-295)

“The results indicate that HAL training might help in correcting the patient’s abnormal gait pattern, leading to gains in motor learning and ensuring consistent signaling of appropriate sensorimotor interactions in the patient’s central nervous system. A previous study reported increased muscle synergies while walking with HAL, suggesting that spinal gait control is altered with HAL [18]. Another study reported that HAL motion assistive technologies could contribute to improvements in walking abilities by facilitating proper joint motion and loading and unloading muscle movements in patients with chronic severe incomplete tetraplegic SCI [26]. The current study conducted HAL training for a patient with non-traumatic spinal cord dysfunction, however, it is likely to observe similar improvements of gait profiles.”

Reviewer 2 Report

Thank you for giving me the possibility to review this interesting case report.  The paper is well written, the data ate clearly reported, discussions are complete.   However, before considering it for publication,  the following concerns should be addressed: 1. Please add preoperative spine MRI and CT images   2. Please detail the surgical strategy    3. Please add an histologic image of the neoplasia   4. Please detail the immediate postoperative care

Author Response

Our point-by-point Responses:

We deeply thank you for taking the time and effort necessary to review our manuscript and provide us with these valuable comments. We have revised our manuscript accordingly. Please note that our changes to the manuscript are highlighted in red font.

Reviewer #2

General comment

Thank you for giving me the possibility to review this interesting case report. The paper is well written, the data ate clearly reported, discussions are complete. However, before considering it for publication, the following concerns should be addressed: 1. Please add preoperative spine MRI and CT images 2. Please detail the surgical strategy 3. Please add an histologic image of the neoplasia 4. Please detail the immediate postoperative care

Response

Thank you for your comments and valuable remarks. We have ensured that all your concerns are addressed in the revised manuscript. Please find our responses to specific comments below.

Specific comment

  1. Please add preoperative spine MRI and CT images

Response

According to your suggestion, we have added the following images to the revised Figure 1 (A, B & C): mid-sagittal T2-weighted MRI of the thoracic spine region (T8) and coronal T2-weighted MRI (T7/8, T8, T8/9), sagittal and coronal computed tomography at the level of T8.

Specific comment

  1. Please detail the surgical strategy

Response

We have included the surgical strategy in detail and modified patient information in the case presentation section.

Revised Manuscript in the case presentation section (Page: 3, lines: 92-102)

Surgical procedure and postoperative course

Surgical treatment involved T7-T8 laminectomy, T9 partial laminectomy, and tumor resection. The dura mater was cut transversely in the ventral direction and the tumor and dura mater were removed in a lump. All surgical procedures were undertaken carefully and accomplished uneventfully. The tumor was diagnosed as a transitional meningioma using histopathology (normal cellularity, sheeting, small cell, prominent nucleoli, spinal cord invasion and few mitosis; World Health Organization classification grade I). The day after surgery, the patient was allowed to sit and walk without an orthosis. Conventional rehabilitation was initiated immediately after permission by the attending physician, since the patient had severe dysfunctional control of voluntary movements and balance and was unable to stand and walk without assistance.

Specific comment

  1. Please add an histologic image of the neoplasia

Response

Unfortunately, we didn’t have an histologic image of the neoplasia this patient. Therefore, we have added information about the diagnostic histopathology report to the revised manuscript. Please refer to the response to Specific comment 2

Specific comment

  1. Please detail the immediate postoperative care

Response

We have added the immediate postoperative care details to the revised manuscript. Please refer to the response to Specific comment 2.

Round 2

Reviewer 1 Report

I am satisfied with the improvements of the article. Authors did all the requested changes.

Reviewer 2 Report

All the raised concerns have been addressed in the revised version of the manuscript. The paper is now suitable of publication.